# Constraining Action Sequences with Formal Languages for Deep Reinforcement Learning

## Abstract

We study the problem of deep reinforcement learning where the agent's action sequences are constrained, e.g., prohibition of dithering or overactuating action sequences that might damage a robot, drone, or other physical device. Our model focuses on constraints that can be described by automata such as DFAs or PDAs. We then propose multiple approaches to augment the state descriptions of the Markov decision process (MDP) with summaries of recent action histories. We empirically evaluate these methods applying DQN to three Atari games, training with reward shaping. We found that our approaches are effective in significantly reducing, and even eliminating, constraint violations while maintaining high reward. We also observed that the total reward achieved by an agent can be highly sensitive to how much the constraints encourage or discourage exploration of potentially effective actions during training, and, in addition to helping ensure safe policies, the use of constraints can enhance exploration during training.

## 1 Introduction

Deep reinforcement learning (DRL) shows great potential in learning decision-making agents with application to many areas, including safety-critical ones such as credential identification, autonomous vehicles, finance, and healthcare. Problematically, the inner workings of DRL models are very difficult to understand, control, or make verifiable claims about. This leads to the problem of how to ensure safe behavior in a learned agent. We propose a method using reward shaping (Ng et al., 1999; Pecka and Svoboda, 2014) and MDP state augmentation to bias the training process away from unsafe behaviors that have been defined in formal languages.

We describe a general framework for incorporating constraints in deep reinforcement learning (DRL) that are expressed as **formal languages** over an agent's action space. We explore several strategies for augmenting the state of the agent so as to maximize rewards subject to constraint satisfaction. To illustrate, consider the **no-1D-dithering** constraint in Figure 1 defined over an Atari Breakout agent with four actions: fire ($f$), no move ($n$), move left ($\ell$), and move right ($r$). The figure's minimized deterministic finite state automaton (DFA) accepts strings ending in $(\ell\,r)^2$ and judges them to be constraint violations. Negative rewards can be applied whenever a violation is detected to drive DRL to avoid that behavior, but the effectiveness of such reward shaping will depend on the ability of the agent to correlate the negative rewards with features of the state. The features that are relevant to formal language constraints are inherently history-based and they may not be present in the original agent state.

A straightforward state augmentation might add a finite action history. For the example property, a history of 3 or more actions would permit the DRL agent to "see" the structure of violating suffixes, but such an approach would not be effective for constraints that restrict sequences like $(\ell\,f^*\,r)^2$, since there may be arbitrarily many $f$ actions in a violating suffix.

Generally speaking, formal languages provide a rich set of representations that can be exploited for augmenting agent state. For a DFA, a natural approach is to monitor the action history recording the current DFA state and adding it to the agent's input state of the MDP. Referring to Figure 1, if the current DFA state were $q_3$, then when an agent takes the $r$-action, DRL has the chance to relate the detected violation with the agent's augmented input state.

While we restrict our attention here to constraints expressed as regular languages defined over agent actions, the approach can be extended to richer formal languages, e.g., pushdown automata where the augmented state includes automata and stack embeddings, and to settings where the MDP state conditions the constraint.

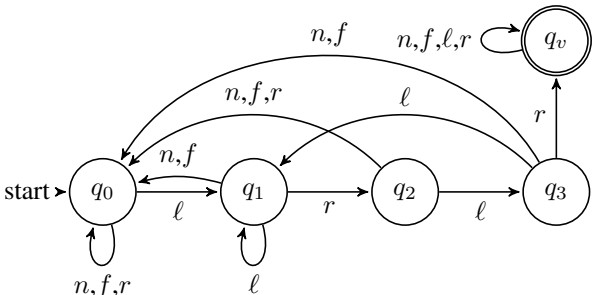

Figure 1: No-1D-dithering constraint: $.^* (\ell\, r)^2$

Our contributions include: (1) a model for constraining RL with respect to action sequences, represented by automata such as DFAs; (2) methods that augment the MDP state descriptions with representations of action sequences in order to learn to avoid constraint violations while maximizing discounted reward; and (3) experimental results comparing our approaches (trained with reward shaping) to a baseline method trained without knowledge of the constraints. We found that our approaches are effective in significantly reducing, and even eliminating, constraint violations while maintaining high reward. We also observed that the total reward achieved by an agent can be highly sensitive to how much the constraints encourage or discourage exploration of potentially effective actions during training. For example, training to avoid violating the no-1D-dithering constraint in the game Breakout significantly negatively impacts the reward the agent receives in game play. In contrast, imposing a generalization of this constraint (no-2D-dithering) when learning Seaquest can **significantly increase reward** during game play. This is because in Breakout, a dithering action is in fact a useful tactic (increasing the effective size of the paddle) whereas in Seaquest, dithering is ineffective, so the constraint during training encourages exploration of alternate actions. We feel that, in addition to helping ensure safe policies, the use of constraints can enhance exploration during training, particularly when training time is too limited to explore effectively.

The rest of our paper is as follows. Section 2 gives related work. Section 3 presents our constrained RL model, and Section 4 describes our approaches. We present our experimental results in Section 5 and conclude in Section 6.

## 2 RELATED WORK

Safety in reinforcement learning is an active research area. The notion of safety during exploration was proposed by Geibel and Wysotzki (2005). It is concerned with the visiting of undesirable states. Some safe exploration methods require *a priori* knowledge of varying degrees. Garcia and Fernandez (2012) utilize a known backup policy to improve on a predefined safe baseline policy while avoiding catastrophic states. Turchetta et al. (2016) expand an initial set of known safe states on regularity assumptions about safety of neighboring states. Others eliminate the need for *a priori* knowledge. Moldovan and Abbeel (2012) emphasize the impossibility of visiting other states from a catastrophic state and consider safety in terms of ergodicity. They optimize $\delta$-safe policies which take the agent to start of exploration with probability at least $\delta$. Krakovna et al. (2018) address ergodicity in dynamic environments and in presence of irreversible actions. Others use a learned a state safety function and a backup policy to revert to a safe state (Hans et al., 2008) or a backup sequence that brings the agent to close proximity of a state discovered to be safe (Mannucci et al., 2018). Lipton et al. (2016) propose a model that remembers previously visited catastrophic states by use of a buffer and a trained safety predictor model. Jansen et al. (2018) use constraint specifications given in probabilistic computation tree logic (PCTL) to assess safety of possible actions for each state. In contrast, our model is neither trained to avoid states nor predicts safe actions given a state. Rather, our model uses state descriptions augmented with information about action history using constraints.

Numerous policy gradient methods that aim at zero-constraint violation during exploration have been proposed. Achiam et al. (2017) proposed a constrained policy optimization method that builds on Trust Region Policy Optimization (TRPO) (Schulman et al., 2015). The policy is kept feasible by using a recovery policy that decreases the constraint value and projects the policy to a feasible set. Chow et al. (2015) also propose a policy optimization method where the objective is a trade-off between return and risk. Lyapunov functions have been used in assessing safety of regions

(Berkenkamp et al., 2017) and guaranteeing global safety of agent's policy. Contrary to the policy optimization works cited so far that consider state-action pair constraints, Dalal et al. (2018) consider constraints that are state-based. Their approach approximates a change in the constraint value of a state based on the previous state and action. In contrast, our constraints are sequence-based and implemented using DFAs instead of scalar-valued functions.

In an agent versus environment game setting, Wen et al. (2015) use linear temporal logic formulas to define permissive strategies. Bou-Ammar et al. (2015) incorporated physical constraints to a safe on-line lifelong multi-task policy gradient learner operating in an adversarial framework. Lee et al. (2018) infer constraints from the actions of a demonstrator on the assumption that states visited by a demonstrator should be safe. Recently, Koller et al. (2018) proposed a Gaussian process (GP)-based RL that can make multi-step ahead safety predictions guarantee safe trajectories that satisfy system constraints. Aoude et al. (2013) propose a real-time GP-based path planning framework that can predict dynamic obstacles. Safe exploration techniques for reinforcement learning have been applied to practical problems. A robotic arm was trained using TRPO and a QP solver (Pham et al., 2017) and by teacher demonstrations where safety is maintained by updating a library of dangerous actions (Martínez et al., 2015). Zhou and Li (2018) incorporate safety specifications expressed in a PCTL formula to the learning phase and use counterexamples to guide policy search in case of PCTL violations in learning from teacher demonstrations. Orseau and Armstrong (2016) consider the cases where an agent might learn to avoid (or seek) human intervention, thus limiting exploration and showing that Q-learning is safely interruptible, i.e., an agent can't learn to avoid (or seek) interruptions. Mhamdi et al. (2017) address the same problem for multi-agent systems in both the joint-action and independent learners' frameworks. Cizelj et al. (2011) and Cizelj and Belta (2012) consider safe vehicle driving in an environment with static and dynamic adversaries using PCTL formulas which describes mission objectives. Held et al. (2017) address the problem of simultaneously maximizing the expected return and managing expected damage for robot operation by adjusting the **unsafety rate** of a policy and the upper limit of physical (state-based) constraints that can incur damage, such as torque. In contrast to state-based constraints, Shen et al. (2018), similar to our model, address action-based constraints in an interactive e-learning setting. However, their approach doesn't directly address constraining action sequences. Rather, they constrain the total number of certain kinds of actions taken. Mazumder et al. (2018) use observed state-action pairs to predict the permissibility of actions given a state. However, their predictor doesn't handle action sequences.

State space augmentation was used by Oh et al. (2016) to predict future frames who applied an attention mechanism to the raw observations (pixel values) and augmented this with past state information retrieved from memory. Dosovitskiy and Koltun (2016) augmented high-dimensional state information with lower-dimensional "measurement information" relevant to the agent's state. In contrast, our model uses direct or encoded information about action history for state augmentation.

## 3  OUR MODEL

We assume that our agent operates in a Markov decision process $(S, A, R, P, \mu)$ with state space $S$, action space $A$, reward function $R : S \times A \times S \to \mathbb{R}$, transition probability function $P : S \times A \times S \to [0, 1]$, and initial state distribution $\mu : S \to [0, 1]$. As usual, its goal is to learn a policy $\pi : S \to A$ to maximize expected discounted cumulative reward from any starting state $s \in S$: $\mathbb{E}\left[ r_t + \gamma r_{t+1} + \gamma^2 r_{t+2} + \cdots \right]$, for discount factor $\gamma < 1$. As with Mnih et al. (2015), we learn $\pi$ via $Q$-learning, though our model can work with other training methods.

The agent's choice of action is restricted by one or more **constraints**. A constraint $C \subset A^*$ is defined as a set of prohibited sequences of actions over $A$. In our work so far, we assume that the set can be described as a regular language, which we represent internally as a DFA[1] $M = (Q, A, \delta, q_0, F)$, where $Q$ is the finite set of the DFA's states, the MDP's action space $A$ serves as $M$'s alphabet, $\delta : Q \times A \to Q$ is the DFA's transition function, $q_0 \in Q$ is the DFA's start state, and $F \subseteq Q$ is the set of accept states. As the agent chooses actions, an external **monitor** steps through $M$ via $\delta$. Whenever an accept state from $F$ is reached, then we say that a **violation** of $C$ has occurred.

---

[1]Ongoing work is to generalize our approaches to handle constraints represented as, e.g., deterministic pushdown automata.

In our model, a constraint can operate in **full reset**, **partial reset**, or **no reset** mode. In full reset mode, at a constraint violation, both the DFA and the MDP are reset, i.e., $M$ resets to initial state $q_0$ and the episode begins anew in a state chosen according to $\mu$. In partial reset mode, the episode does not reset, but the DFA does. In no reset mode, neither the episode nor the DFA resets. In all three modes, a finite punishment is applied whose magnitude correlates with the severity of the violation.

In our learning model, we assume that each constraint is made available in some form to the learner by a benevolent teacher (e.g., a software developer with direct knowledge of the agent's constraints). In our approaches, this information is provided to the agent as a function $f(\cdot)$ applied to the agent's action sequence. In two of our methods, $f(\cdot)$ provides specific context of the agent's action history with respect to the DFA describing the constraint. This allows us to use this context to augment the state description.

# 4    OUR METHODS

All four of our methods use reward shaping (Ng et al., 1999) to train agents in our model to deter constraint violations. Our first approach (***No Augmentation***) directly applies reward shaping, but does not otherwise modify the learning problem (i.e., using $f(\cdot) = \emptyset$ below). Our remaining three methods enable an agent to track its action history to help avoid constraint violations. We explore three approaches of augmenting the MDP state $s_t \in S$. In each approach, at time step $t$ the agent receives $\langle s_t, f(\mathbf{a}_t) \rangle$, a concatenation of current MDP state $s_t$ and some function $f(\cdot)$ evaluated on action sequence $\mathbf{a}_t = \langle a_1, a_2, \ldots, a_t \rangle \in A^*$. We consider three forms of $f(\cdot)$:

1. ***Size-$k$ Action History:*** $f(\mathbf{a}) = \langle a_{t-k+1}, a_{t-k+2}, \ldots, a_{t-1}, a_t \rangle$, where $a_t$ is a one-hot encoding of the action taken at time $t$. In our experiments, we set $k = 10$.

2. ***DFA State One-Hot Encoding:*** $f(\mathbf{a}) = \mathbf{u} \in \{0,1\}^{|Q|}$, where $u_i = 1$ if the monitor indicates that the DFA is in state $q_i \in Q$ and 0 otherwise. $\mathbf{u}$ is a one-hot encoding of the current DFA state.

3. ***DFA State Embedding:*** $f(\mathbf{a}) = \mathbf{v} \in \mathbb{R}^d$, where $\mathbf{v}$ is the embedding of DFA state $q_i$ learned by node2vec (Grover and Leskovec, 2016). Node2vec trains its embeddings via random walks started at each node, allowing the embeddings to represent each node's **context** (e.g., neighborhood or reachability). In our experiments, we set $d = 3$ and performed 200 iterations of 200 random walks of each DFA, each of length 80. Our context window size was 5, and our random walks were a combination of BFS and DFS, setting $p = q = 1$.

Our approaches are meant to learn policies that try to avoid constraint violations. If violations at run time must absolutely be avoided, then we ensure this by modifying the learned policy $\pi$ by a **constraint enforcer** that guarantees constraint compliance as follows. If the policy's chosen action $a$ does not violate the constraint, then the enforcer allows $a$ to be executed in the environment. Otherwise (if $a$ would cause a violation), then the enforcer asks $\pi$ for its next most preferred action, and checks it. This process repeats until $\pi$'s chosen action $a'$ does not violate the

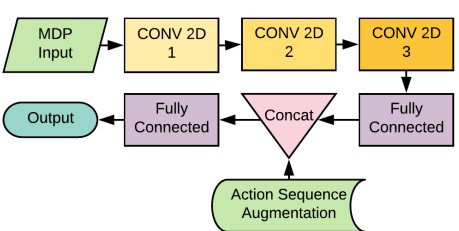

Figure 2: Architecture used in experiments.

constraint. The enforcer then executes $a'$ in the environment. In our experiments, we evaluate all our methods with and without the use of the enforcer.

The architecture we used to implement our methods is the same as in Mnih et al. (2015), with an additional input for augmenting the MDP state with action history, as summarized in Figure 2 and detailed in Figure 6 in the Appendix. We concatenate this action history (denoted "Action Sequence Augmentation" in Figure 2) to the penultimate fully connected layer. We used ReLU activation for all layers except the final fully-connected layer. The final fully-connected layer outputs the $Q$ values.

## 5    EXPERIMENTS

### 5.1    EXPERIMENTAL SETUP

In our experiments, we focused on three Atari games: Breakout, Space Invaders, and Seaquest (Mnih et al., 2015), which have diverse action spaces and high-reward game play strategies. For each game we considered two generic constraints. Each constraint is applicable to each of the games. However, depending on the game, each constraint either coincides or conflicts with effective play strategy. We express these constraints below as regular expressions[2]. These constraints were chosen to reflect ones that might be imposed on a real physical system, e.g., to avoid excessive stress on a robot arm.

1. **No-dithering:** In general, a no-dithering constraint prohibits movements in small, tight patterns that cover very small areas. In one dimension, we define **no-1D-dithering** as having a violation as

$$.^* (\ell\, r)^2$$

   i.e., never move left then right then left then right; here ".*" is any sequence over actions from $A$. In games with larger action spaces such as Seaquest, we generalize this to **no-2D-dithering**, which extends to vertical and diagonal moves and constrains actions that take the agent back to where it started in at most four steps.[3]

2. **No-overactuating:** In general, a no-overactuating constraint prohibits repeated movements in the same direction over a long period of time. In one dimension, a violation is

$$.^* (\ell^4 \cup r^4)$$

   i.e., never move left four times in a row or right four times in a row. In two dimensions, this is extended to include vertical move groups: $.^* (L^4 \cup R^4 \cup U^4 \cup D^4)$. Each of left ($L$), right ($R$), up ($U$) and down ($D$) groups contains the primary direction it's named after and diagonal moves that contain the primary direction, e.g., $L = \ell \cup \ell{+}u \cup \ell{+}d$, where "$\ell{+}u$" is the atomic left-up diagonal action.

For each game/constraint pair, we evaluated the four approaches from Section 4. As a baseline, we also trained a model with no knowledge of the constraints during training (i.e., standard DQN training for the Atari games), but tested with the constraints present. In all experiments, the learned policies were evaluated in **partial reset** mode, reporting total reward and total number of violations per episode.

Our loss function and optimizer were mean absolute error and Adam with a learning rate of $2.5 \times 10^{-4}$. Our discount future reward $\gamma = 0.99$ and we used $\epsilon$-greedy as an exploration policy with $\epsilon$ decreasing from 1 to 0.1 over 10M steps. We set the experience replay memory limit to 1M samples. Every state of the game is represented with 4 frames. We trained the network with random mini-batches of size 32 from the replay memory in every 4 steps. The target network is updated every 10K steps. For each approach of Section 4 and our baseline methods, we trained the agent for 10M steps for 10 different training seeds. We tested each of our 10 trained agents for 10 different test seeds. Each test case includes 100 game episodes running to completion. Our reward shaping signal was $-1000$.

### 5.2    EXPERIMENTAL RESULTS

Figures 3, 4, and 5 present average number of violations and average reward (reward only; no shaping) per test episode. In each plot, "*baseline*" denotes performance of an agent trained without any state augmentation or reward shaping, i.e., the same approach as Mnih et al. (2015). "*No augmentation*" denotes an agent trained with reward shaping but no state augmentation. "*Action history*" denotes augmenting the state with a one-hot encoding of the previous $k = 10$ actions. "*DFA one-hot*" denotes augmenting the state with a one-hot encoding of the current DFA state. Finally, "*dfa embedding*" denotes augmenting the state with a node2vec-learned embedding of the current DFA state.

In each figure, vertical and horizontal error bars indicate 95% confidence intervals. Each symbol without a circle around it denotes performance when the learned policy is tested as-is. A symbol

---

[2]For Seaquest and Space Invaders, movement actions with and without fire are treated as identical.
[3]The regex describing this constraint is 7320 characters long and is thus omitted.

with a circle around it denotes performance when the learned policy $\pi$ was modified by a constraint enforcer as described in Section 4.

Figure 3 shows test results on the game Breakout using the no-dithering and the no-overactuating constraints. The no-dithering constraint can be easily complied with, since all approaches except *baseline* are effective at reducing constraint violations, and all of them get nearly zero violations. Further, we see that simply training without any shaping (*baseline*) and then applying the enforcer greatly outperforms everything else in terms of reward. Similar things can be said of the no-overactuating constraint, except compliance with it is more difficult to learn: every approach save *action history* has several violations (though significantly fewer than *baseline*), and the enforcer reduces reward more significantly for *baseline*. However, *baseline* with the enforcer still outperforms everything else with both constraints. We attribute this to both constraints being antithetical to good game play: dithering increases Breakout's effective paddle size and overactuation allows more flexibility in how the paddle can be moved into position to hit the ball. Since the shaping signal applied during training discouraged these actions, exploration was inhibited and poorly performing non-violating actions were learned by our methods.

Figure 4 shows test results on the game Space Invaders using the no-dithering and no-overactuating constraints. We see that the no-dithering constraint is difficult to learn to completely comply with, since every method had a positive number of violations without the enforcer's help. While none of the methods eliminated violations completely, *dfa one-hot* and *dfa embedding* were able to reduce violations to less than 10% of that of *baseline*. We also note that *baseline*, *no augmentation*, and *dfa embedding* each took significant hits in reward when the enforcer was applied. In contrast, *action history* and (particularly) *dfa one-hot* saw little drop in reward when the enforcer was applied, so their policies' first non-violating choices were on average nearly as good as their top (violating) actions when a violation was imminent. This suggests that they learned to avoid situations in which a violating action is the only quality option that it has.

Space Invaders with no-overactuation is a different case, where all approaches reduced violations to less than 20% of that of *baseline* and application of the enforcer slightly **improved** reward for all approaches, save *dfa embedding*. This is an example of a constraint that is not antithetical to good game play, and in fact seems to encourage exploration during training.

Figure 5 shows test results on the game Seaquest using the no-2D-dithering and no-overactuation constraints. These results are similar to those with Space Invaders no-overactuation in that application of the enforcer can increase reward, especially with *no augmentation*, *action history*, and *dfa embedding*. A key difference between this game and the other ones is that *baseline* had significantly lower reward than some other approaches, particularly *dfa one-hot*. The use of these constraints during training seems to encourage exploration of better policies. For example, moving in a tight pattern and ending up where one started would not be a good policy for game play. Since this is prohibited by no-2D-dithering, arguably this constraint is forcing the agent to explore alternatives, and hence discover better policies.

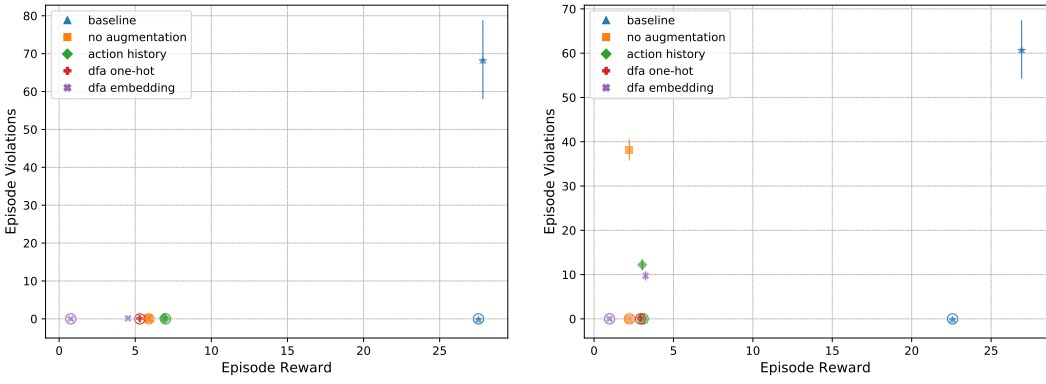

Figure 3: Results for Breakout: (left) no-1D-dithering; (right) no-overactuating.

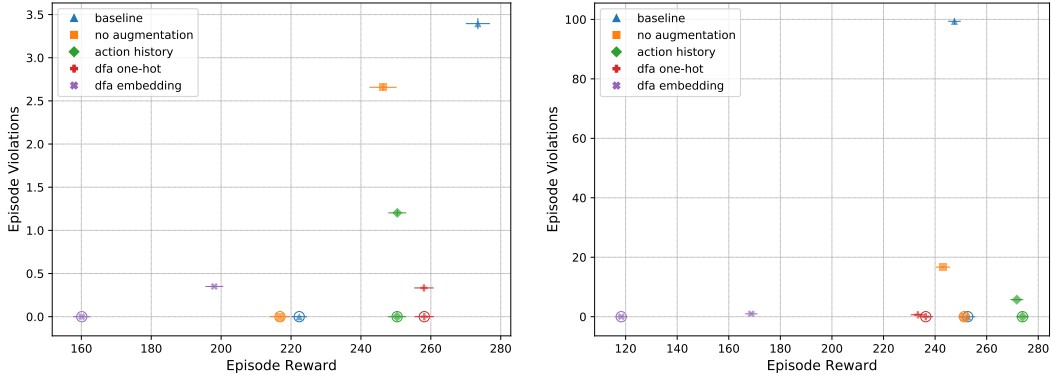

Figure 4: Results for Space Invaders: (left) no-1D-dithering; (right) no-overactuation.

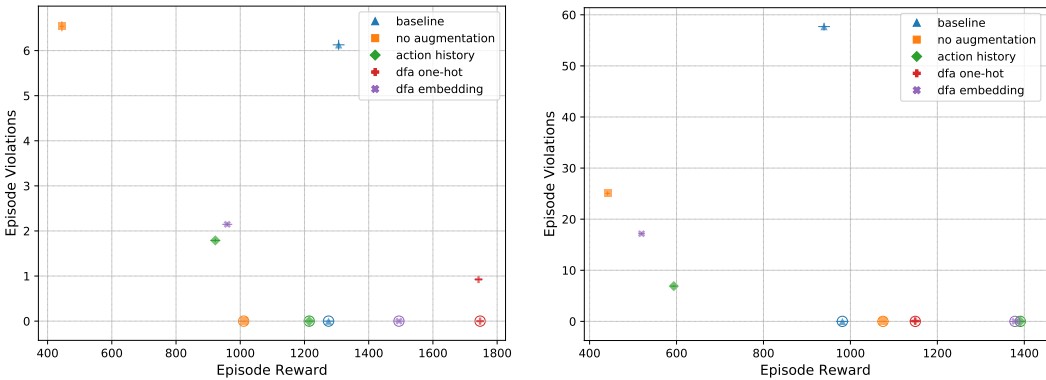

Figure 5: Results for Seaquest: (left) no-2D-dithering; (right) no-overactuation.

### 5.3 DISCUSSION

We draw two broad conclusions from our experimental results, which can be described in the dimensions of **violations** and of **rewards**. First, the ability to learn to avoid constraint violations without the enforcer's help unsurprisingly depends on the complexity of the constraint. Among our approaches, *dfa one-hot* was overall the most effective in minimizing violations, followed by (in most cases) *dfa embedding* and then *action history*.

The second dimension of analysis regards rewards. Specifically, among the violation-free policies (particularly those where the enforcer was applied), the reward achieved was closely related with how the constraint conflicts with good game play. When a constraint discouraged actions that would normally yield higher reward (both constraints in Breakout), we saw that training without any constraint knowledge (*baseline*) and then applying the enforcer returned the most reward. However, when a constraint discouraged unhelpful actions (both constraints in Seaquest), exploration was encouraged during training and testing reward was high, even with the enforcer in place. Further, when the agent did not learn to avoid constraint violations very well, applying the enforcer significantly increased the total reward (*no augmentation, action history*, and *dfa embedding* on both constraints in Seaquest).

These conclusions suggest that careful consideration is necessary when training agents subject to such constraints. In particular, one should consider how the constraint(s) align with the actions of effective policies. If they would inhibit the learning of a good policy, then care is needed during training, e.g., via delaying the application of the shaping signal (or increasing it over time), or by a hybrid training approach that combines our methods with the unconstrained *baseline* approach. Further, whether or not safety-based constraints apply to the learning problem, one should consider

using constraints to focus exploration during training away from actions that are known to be less effective, i.e., use constraints to encode human-defined game play tactics. This could be particularly helpful in cases where training time is too limited for the agent to consider less useful actions.

## 6    CONCLUSION

We proposed a model for constraining RL by using reward shaping and MDP state augmentation to bias the agents to avoid unsafe action sequences that are defined by formal languages. We applied our methods to three Atari games with very different action spaces and performed a thorough empirical study. Our findings revealed that all four approaches are effective in reducing or even eliminating constraint violations while maintaining and at times increasing reward. Furthermore, we found that constraints can not only be used to avoid unsafe behaviors, but also be defined to encourage exploration to aid agents to learn faster.

One major avenue of future work is to better characterize the relationships between constraints (defined as formal languages) and effective policies (defined as functions from states to actions). A thorough understanding of such relationships will inform the definition of constraints and the learning approaches used to optimize the policies. It will also help in defining constraints to enhance exploration during training.

There are many more avenues of future work for this project. First, in Section 3 we define a constraint $C$ to be a subset of $A^*$, i.e., sequences of actions. It would be natural to instead define $C \subset A^* \times S^*$, so the constraints can restrict actions only in certain MDP states, if desired. One way to keep the sizes of the DFAs manageable in this case is to define $C \subset A^* \times S'^*$, where $S'$ is an alphabet where distinct symbols describe distinct **equivalence classes** of states in $S$. E.g., symbol $m \in S'$ might denote all states in $S$ where a mother ship appears in Space Invaders. It would also be interesting to describe constraints via, e.g., deterministic pushdown automata rather than DFAs.

The *dfa embedding* approach did not perform as well as we thought that it would. It is possible that this can be improved by modifying the random walk approach to be more appropriate for this task, e.g., to incorporate how close a state is to an accept state.

Application of the constraint enforcer during game play eliminated constraint violations, but the reward of the adjusted policies depended on the quality of the first non-violating choices that the policies made. Future work is to incorporate the enforcer into training to try to increase the expected reward of not only the top choice, but also the top choice that is compliant.

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

## A  ARCHITECTURE DETAILS

Figure 6 presents a detailed view of our architecture. Part of our architecture is the same as in Mnih et al. (2015), using a convolutional stack that processes four $84 \times 84$ frames at a time. To this stack we add an input layer for augmenting DFA state.

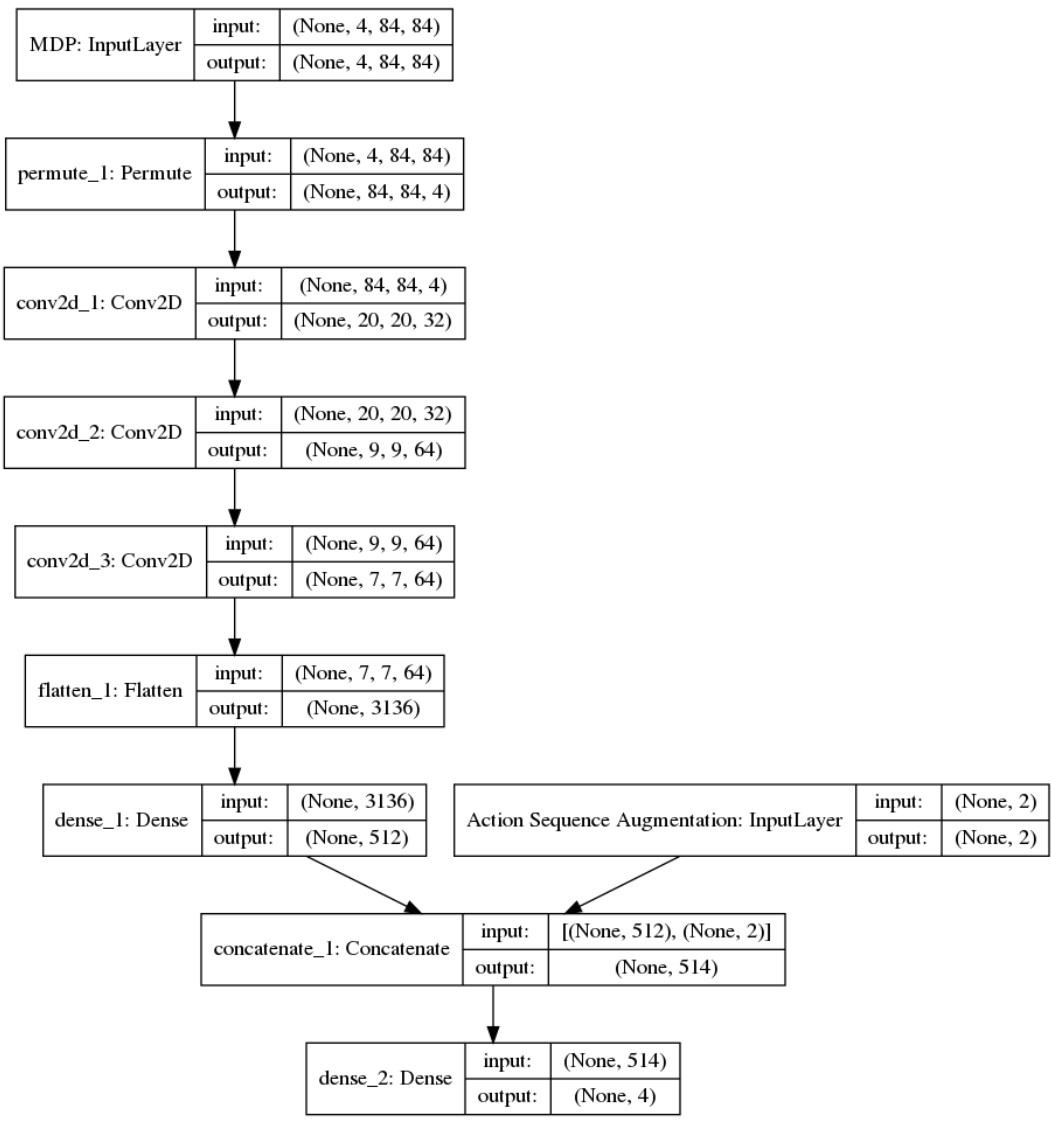

Figure 6: Architecture used in our experiments.

