# OpenReview forum: "Constraining Action Sequences with Formal Languages for Deep Reinforcement Learning"
_ICLR.cc/2019/Conference_

### Official Review · AnonReviewer2 · 2018-11-02

**Rating:** 4
**Confidence:** 3

**Review:**

This work aims to use formal languages to add a reward shaping signal in the form of a penalty on the system when constraints are violated. There is also an interesting notion of using an embedding based on the action history to aid the agent in avoiding violations. However, I do not believe this paper did a good enough job in situating this work in the context of prior work — in particular (Camacho 2017). There is a significant related work section that does an ok job of describing many other works, but to my knowledge (Camacho 2017) is the most similar to this one (minus the embedding), yet is not mentioned here. It is difficult to find all related work of course, so I would encourage revision with detailed description of the novelty of this work in comparison with that one. I would also encourage an more thoughtful examination of the theoretical ramifications of the reward shaping signal with respect to the optimal policy as (Camacho 2017) do and as is modeled in the (Ng 1999) paper. As of this revision, however, I'm not sure I would recommend it for publication. Additionally, I suggest that the authors describe the reward shaping mechanism a bit more formally, it was unclear whether it fits into Ng's potential function methodology at first pass.

Comments:

+ It would be nice to explain to the reader in intuitive terms what “no-1D-dithering” means near this text. I understand that later on this is explained, but for clarity it would be good to have a short explanation during the first mentioning of this term as well.
+ It would be good to clarify in Figure 1 what . * (lr)^2 is since in the main text near the figure is is just (lr)^2 and the .* is only explained several pages ahead
+ An interesting connection that might be made is that Ng et al.’s reward shaping mechanism, if the shaping function is based on a state-dependent potential then the optimal policy under the new MDP is still optimal for the old MDP. It would be interesting to see how well this holds under this holds under this schema. In fact, this seems like analysis that several other works have done for a very similar problem (see below).
+ I have concerns about the novelty of this method. It seems rather similar to

Camacho, Alberto, Oscar Chen, Scott Sanner, and Sheila A. McIlraith. "Decision-making with non-markovian rewards: From LTL to automata-based reward shaping." In Proceedings of the Multi-disciplinary Conference on Reinforcement Learning and Decision Making (RLDM), pp. 279-283. 2017.
Camacho, Alberto, Oscar Chen, Scott Sanner, and Sheila A. McIlraith. "Non-Markovian Rewards Expressed in LTL: Guiding Search Via Reward Shaping." In Proceedings of the Tenth International Symposium on Combinatorial Search (SoCS), pp. 159-160. 2017.

However, that work proposes a similar framework in a much more formal way. In fact, in that work also a DFA is used as a reward shaping signal -- from what I can tell for the same purpose through a similar mechanism. It is possible, however, that I missed something which contrasts the two works.

Another work that can be referenced:

De Giacomo, Giuseppe, Luca Iocchi, Marco Favorito, and Fabio Patrizi. "Reinforcement Learning for LTLf/LDLf Goals." arXiv preprint arXiv:1807.06333 (2018).

I think it is particularly important to situate this work within the context of those others.

+ General the structure of the paper was a bit all over the place, crucial details were spread throughout and it took me a couple of passes to put things together. For example, it wasn't quite clear what the reward shaping mechanism was until I saw the -1000 and then had to go back to figure out that basically -1000 is added to the reward if the constraint is violated. I would suggest putting relevant details all in one place. For example, "Our reward shaping function F(x) was  { -1000, constraint violation, 0 otherwise}".

---

### Official Review · AnonReviewer3 · 2018-11-02
**Approach for biasing RL agent away from particular action sequences**

**Rating:** 3
**Confidence:** 4

**Review:**

This paper presents an approach for biasing an agent to avoid particular action sequences. These action sequence constraints are defined with a deterministic finite state automaton (DFA). The agent is given an additional shaping reward that penalizes it for violating these constraints. To make this an easier learning problem for the agent, its state is augmented with additional information: either an action history, the state of the DFA, or an embedding of the DFA state. The authors show that these approaches do reduce these action constraint violations over not doing anything about them.

It's unclear to me what the use case is for constraints solely on the action space of the agent, and why it would be useful to treat them this way. The authors motivate and demonstrate these constraints on 3 Atari games, but it is clear that the constraints they come up with negatively affect performance on most of the games, so they are not improving performance or safety of the agent. Are there useful constraints that only need to view the sequence of actions of the agent and not any of the state?  If there are such constraints, why not simply restrict the agent to only take the valid actions? What is the benefit of only biasing it to avoid violating those constraints with a shaping reward? This restriction was applied during testing, but not during training.

In all but the first task (no 1-d dithering in breakout), none of the proposed approaches were able to completely eliminate constraint violations. Why was this? If these are really constraints on the action sequence, isn't this showing that the algorithm does not work for the problem you are trying to solve?

The shaping reward used for the four Atari games is -1000. In most work on DQN in Atari, the game rewards are clipped to be between -1 and 1 to improve stability of the learning algorithm. Were the Atari rewards clipped or unclipped in this case? Did having the shaping reward be such large magnitude have any adverse effects on learning performance?

Adding a shaping reward for some desired behavior of an agent is straightforward. The more novel part of this work is in augmenting the state of the agent with the state of a DFA that is tracking the action sequence for constraint violations. Three approaches are compared and it does appear that DFA one-hot is better than the other approaches or no augmentation.

Pros:
- Augmenting agent state with state of DFA tracking action sequence constraints is novel and useful for this problem
Cons:
- Unclear if constraints on action sequences alone useful
- No clear benefit of addressing this problem through shaping rewards.
- No comparison to simply training with only non-violating action sequences.
- Algorithm still results in action constraint violations in 5/6 tasks.

---

### Official Review · AnonReviewer1 · 2018-11-06
**interesting approach with inconclusive results**

**Rating:** 5
**Confidence:** 4

**Review:**

This paper presents an DFA-based approach to constrain certain behavior of RL agents, where "behavior" is defined by a sequence of actions. This approach assumes that the developer has knowledge of what are good/bad behavior for a specific task and that the behavior can be checked by hand-coded DFAs or PDAs. During training, whenever such behavior is detected, the agent is given a negative reward, and the RL state is augmented with the DFA state. The authors experimented with different state augmentation methods (e.g. one-hot encoding, learned embedding) on 3 Atari tasks.

The paper is clearly written. I also like the general direction of biasing the agent's exploration away from undesirable regions (or conversely, towards desired regions) with prior knowledge. However, I find the results hard to read.

1. Goal. The goal of this work is unclear. Is it to avoid disastrous states during exploration / training, or to inject prior knowledge into the agent to speed up learning, or to balance trade-offs between constraint violation and reward optimization? It seems the authors are trying to do a bit of everything, but then the evaluation is insufficient. For example, when there are trade-offs between violation and rewards, we expect to see trade-off curves instead of single points for comparison. Without the trade-off, I suppose adding the constraint should speed up learning, in which case learning curves should be shown.

2. Interpreting the results. 1) What is the reward function used? I suppose the penalty should have a large effect on the results, which can be tuned to generate a trade-off curve. 2) Why not try to add the enforcer during training? A slightly more complex baseline would be to enforce with probability (1-\epsilon) to control the trade-off. 3) Except for Fig 3 right and Fig 4 left, the constraints doesn't seem to affect the results much (judging from the results of vanilla DQN and DQN+enforcer) - are these the best settings to test the approach?

Overall, an interesting and novel idea, but results are a bit lacking.

---

### Author Response · Authors · 2018-11-26
**We appreciate the detailed reviews**

We thank the reviewers for their detailed comments, and we are using this feedback to improve the paper.

---

### Meta-Review · Area_Chair1 · 2018-12-14
**The paper needs to be improved**

**Confidence:** 4
**Recommendation:** Reject

**Metareview:**

The paper studies the problem of reinforcement learning under certain constraints on action sequences. The reviewers raised important concerns regarding (1) the general motivation, (2) the particular formulation of constraints in terms of action sequences and (3) the relevance and significance of experimental results. The authors did not submit a rebuttal. Given the concerns raised by the reviewers, I encourage the authors to improve the paper to possibly resubmit to another venue.